# Impact of Adverse Childhood Experiences (ACE) and Childhood Protective Factors (CPF) on physical and mental health of medical students of a public sector medical university

Usman Ali ⓘ*, Nazish Imran, Muhammad Shaheryar Khan

Academic Department of Psychiatry and Behavioral Sciences, King Edward Medical University, Lahore, Pakistan

* econtactusman@gmail.com

## Abstract

The current study aimed to assess the impact of Adverse Childhood Experiences (ACE) and Childhood Protective Factors (CPF) on physical and mental health of medical students of a public sector medical university. An online cross-sectional descriptive study was carried out at a public sector medical university in Lahore, Pakistan from 1 June 2024 to 30 August 2024. The study tool consisted of sociodemographic questions, an Adverse Childhood Experience (ACE) questionnaire, and a Resilience Questionnaire for measuring childhood protective factors and self-reported physical and mental health in the last 4 weeks. Data was entered and analyzed using Statistical Package for Social Sciences (SPSS) Version 26.0. A total of 368 medical students participated in the survey of which 125 (33.96%) were male and 243 (66.03%) were females. A higher Childhood Protective Factor (CPF) score significantly predicted good physical health (AOR 1.516, 95% CI 1.085–2.120, p <.01) and excellent physical health (AOR 1.61, 95% CI 1.151–2.263, p <.001). A higher ACE score significantly predicted fair mental health (AOR .767, 95% CI .624–.944, p <.05), good mental health (AOR .746, 95% CI .604–.922, p <.01), and excellent mental health (AOR .746, 95% CI .604–.922, p <.01). Our study shows that ACE was a significant predictor of self-reported mental health but not physical health. However, CPF was a significant predictor of physical and mental health.

## Introduction

The early life period from childhood to adolescence is marked by numerous physiological and psychological changes in an individual [1]. The period is marked by the development of personality, attitudes and perceptions towards oneself and others, the development of self-esteem, and the setting of life aspirations and ambitions. During this period various physiological and genetic mechanisms play their role in neurological development which goes parallel with psychological development and emotional maturity [1]. Consequently, any adversity, either experienced or witnessed by a person may have long-term consequences in the person's life [1–4].

**Data availability statement:** Available at GITHUB: https://github.com/Usmanali1994/ACEI.

**Funding:** The authors received no specific funding for this work.

**Competing interests:** The authors have declared that no competing interests exist.

There is a great body of evidence supporting that adverse childhood events can predict poor physical and mental health as well as low odds of completing education, higher probability of poor socioeconomic status, and so on [3,4]. Hughes et al., have published a meta-analysis to find out the effect of childhood adversities on various health outcomes. According to the results of this meta-analysis, adverse childhood experiences were strongly linked with violence either directed to oneself or towards others, risk-taking sexual behaviors, problematic use of alcohol, and poor mental health. A recently published population-based study from the United States of America states that experiencing childhood physical, verbal, or sexual abuse, witnessing parental domestic violence, experiencing parental divorce, and living with someone who was depressed, used alcohol or recreational drugs, or spent time as jail inmate was linked to poor self-rated physical health, limited functionality as an adult, and diabetes or cardiovascular disease as an adult [1]. The underlying mechanisms by which adverse childhood events lead to negative outcomes remain incompletely understood but are believed to be multifactorial. These experiences can have long-term detrimental effects on a person's life, potentially through immune or stress-mediated injury to the nervous system, poor self-esteem, behavioral maladaptation, or indirectly through poor education and socioeconomic status—factors that independently influence health in adulthood [1].

Alongside there are a number of positive factors which may buffer the effect of childhood adversities and predict better health outcomes and quality of life [1,5–7]. These factors include positive emotional support from parents and teachers, trusting relationships, and an enabling environment that facilitates the self-expression of children. Consequently, a key public health strategy is to strengthen these protective factors while reducing childhood adversities.

According to Pakistan Bureau of Statistics, Pakistan is the fifth most populous country in the world. The country is ranked among low-middle-income countries, with a low human development index and lower access to healthcare [8]. Parenting is often done in conventional manners which may include corporal punishment. Children may observe intimate partner violence during developmental age [9]. As highlighted above these factors can have a negative impact on the well-being of a person in adulthood. On the opposite side, Pakistan still has a strong family structure where children are looked after by parents as well as by extended family [9].

There is a scarcity of data on adverse childhood experiences among medical students, who may have additional stress related to career and educational demands [10]. Medical students are in their late teenage or early adulthood and thus face unique challenges. Adverse childhood experiences may have detrimental effects on their physical and mental health and thus may hamper the realization and achievement of their true potential. The household environment may be a point of intervention that mitigates childhood adversities, thus there is a need to expand knowledge to the base to better understand the effects of Adverse Childhood Experiences (ACE) and Childhood Protective Factors (CPF) on the physical and mental health of medical students. To the best of our knowledge, no study on adverse childhood experiences and childhood protective factors and their relation with health outcomes has been conducted in Pakistan. The study aims to assess the impact of Adverse Childhood Experiences (ACE) and Childhood Protective Factors (CPF) on the physical and mental health of medical students of a public sector medical university.

## Materials and methods

Ethics Statement: Ethical approval was taken from the Institutional Review Board of King Edward Medical University, Lahore, Pakistan (Letter no. 318/RC/KEMU dated 18 May 2024).

A cross-sectional descriptive study was carried out at King Edward Medical University from 1 June 2024 to 30 August 2024. Medical students over eighteen years of age who

enrolled at King Edward Medical University were included in the study. Written consent was taken before inclusion in the study. The study tool consisted of sociodemographic questions, Adverse Childhood Experience (ACE) questionnaire, Resilience Questionnaire, and self-reported physical and mental health in the last 4 weeks reported on the Likert scale. Adverse Childhood Experience is a validated questionnaire consisting of ten items asking the subject about potential traumatic experiences before eighteen years of age. Five of the items on the ACE questionnaire are related to childhood abuse (physical abuse, mental abuse, emotional abuse, or childhood neglect). The other five items are related to observing or experiencing potentially adverse events, i.e., an alcoholic parent, a family member with mental illness or in jail, and the disappearance of a parent through divorce, death, or abandonment. Answering yes on any item on the scale is assigned a score of one. A cumulative score can range from 0 to 10 with a higher score corresponding to more experiences of childhood adversity [10]. Childhood protective factors were assessed using the Resilience questionnaire. It is a 14-item questionnaire with responses marked on five possible answers (definitely true, probably true, not sure, probably not true, definitely not true). The questionnaire was developed by the early childhood service providers including clinicians, mental health experts, at the Southern Kennebec Healthy Start, Augusta, Maine. The questionnaire is based on literature review and was modeled on the same scoring systems as ACE Study questions. Scores can be calculated by assigning 1 point to each statement answered "definitively true" or "probably true" and 0 points to "not sure," "probably not true," and "definitively not true" responses. The score can range from 0 to 14 with a higher score showing more childhood protective factors (CPF) [10].

Descriptive statistics (frequencies and percentages) were computed as well as mean scores for adverse childhood experience and resilience scale. Comparison between male and female participants was done using chi-square for categorical variable and Mann Whitney U test for continuous variable after ascertaining the distribution of data. A bivariate correlation was computed between self-reported physical and mental health in the last 4 weeks with sociodemographic variables, ACE score, and resilience questionnaire score. Variables with a correlation coefficient of 0.1 with self-reported physical and mental health were included in multinomial logistic regression. Assumptions for multinomial logistics regression were checked as described in the literature [11]. Data was entered and analyzed using Statistical Package for Social Sciences (SPSS) Version 26.0.

## Results

A total of 368 medical students participated in the survey of which 125 (33.96%) were male and 243 (66.03%) were females. Among sociodemographic characteristics, there was a significant age difference in median ranks between males and females. There was a significant difference between the employment status of fathers with higher proportions of fathers of male medical students being unemployed compared with the fathers of female medical students. There was no statistical difference between ACE scores, CPF scores, and self-reported physical and mental health of male and female medical students. Table 1 compares the sociodemographic characteristics of male and female medical students.

Around 89.5% of medical students had an ACE score of 3 or less, 86.4% of males and 91.4% of females had an ACE of 3 or less than 3. We compared ACE scores for individual items for each category between male and female medical students using the chi-square test as shown in Table 2. There was a statistically significant difference between male and female medical students, in terms of sexual abuse, with 18.4% of male medical students facing sexual abuse compared with 11.1% of female students. Similarly, 24.0% of male medical students stated that they lived with a household member with mental illness while 13.6% of female medical

**Table 1. Sociodemographic characteristics of participants (n = 368).**

| Variable | Category | Total n (%) | Male n (%) | Female n (%) | P value |
|---|---|---|---|---|---|
| Age in years (mean rank) | | 20.64 | 192.52 | 169.60 | .040* |
| Mothers' education | | | | | .01** |
| | Not educated | 17 (4.61%) | 12 (9.76%) | 5 (2.07%) | |
| | Educated up to 10th grade or religious education | 60 (16.30%) | 26 (21.14%) | 34 (14.05%) | |
| | Educated above matric up to graduation | 245 (66.58%) | 68 (55.28%) | 177 (73.14%) | |
| | Higher education | 40 (10.87%) | 17 (13.82%) | 26 (10.74%) | |
| Mothers' occupation | | | | | .817 |
| | Housewife | 266 (72.28%) | 88 (77.19%) | 178 (76.07%) | |
| | Working woman | 82 (22.28%) | 26 (22.18%) | 56 (23.93%) | |
| Fathers' education | | | | | .204 |
| | Not educated | 1 (0.27%) | 1 (1.59%) | 0 (0.00%) | |
| | Educated up to 10th grade or religious education | 78 (21.20%) | 29 (46.03%) | 49 (39.20%) | |
| | Educated up to graduation | 50 (13.59%) | 12 (19.05%) | 38 (30.40%) | |
| | Higher education | 59 (16.03%) | 21 (33.33%) | 38 (30.40%) | |
| Fathers' occupation | | | | | .006** |
| | Employed | 262 (71.20%) | 82 (83.67%) | 180 (93.75%) | |
| | Unemployed | 28 (7.61%) | 16 (16.33%) | 12 (6.25%) | |
| Household income | | | | | .259 |
| | 50,000 PKR or less | 23 (6.25%) | 8 (9.09%) | 15 (9.20%) | |
| | 50,001 PKR to 100,000 PKR | 55 (14.95) | 25 (28.41%) | 30 (18.40%) | |
| | 100,001 PKR to 200,000 PKR | 93 (25.27%) | 32 (36.36%) | 61 (37.42%) | |
| | 200,001 PKR or more | 80 (21.74%) | 23 (26.14%) | 57 (34.97%) | |
| ACE score (mean rank) | | – | 194.34 | 179.44 | .178 |
| CPF score (mean rank) | | – | 174.34 | 189.73 | .185 |
| Self-reported physical health | | | | | .141 |
| | Very poor | 34 (9.24%) | 14 (11.38%) | 20 (8.33%) | |
| | Poor | 61 (16.58%) | 22 (17.89%) | 39 (16.25%) | |
| | Good | 196 (53.26%) | 71 (57.72%) | 125 (52.08%) | |
| | Very good | 41 (11.14%) | 7 (5.69%) | 34 (14.17%) | |
| | Excellent | 31 (8.42%) | 9 (7.32%) | 22 (9.17%) | |
| Self-reported mental health | | | | | .151 |
| | Very poor | 31 (8.42%) | 13 (10.66%) | 18 (7.53%) | |
| | Poor | 85 (23.10%) | 34 (27.87%) | 51 (21.34%) | |
| | Fair | 110 (29.89%) | 39 (31.97%) | 71 (29.71%) | |
| | Good | 83 (22.55%) | 25 (20.49%) | 58 (24.27%) | |
| | Excellent | 52 (14.13%) | 11 (9.02%) | 41 (17.15%) | |

**Note:** *. =p <.05 level. **. =p<.01 level. ***. p = <.001 level.

students stated that they lived with a household member with mental illness ($\chi2=6.50$ (1) p-value <.01).

We conducted multinomial logistic regression for self-reported physical and mental health. Results of multinomial regression for predictors of self-reported physical health are given in Table 3. The model included ACE score, CPF score, and sex as predictors for physical health. Overall, the model was a significant fit for predicting self-reported physical health ($\chi2(24)$ = 45.832, p-value, <.001) and explained some degree of variance ($R^2$=.119). Higher CPF score

**Table 2. Comparison of ACE scores between male and female medical students.**

| Serial no. | ACE question | Total n (%) | Males who responded yes n (%) | Females who responded yes n (%) | Chi-square χ2(df) p-value |
|---|---|---|---|---|---|
| 1. | Emotional abuse | 99 (26.90%) | 36 (28.8%) | 63 (25.90%) | 3.87 (1), p =.541 |
| 2. | Physical abuse | 64 (17.39%) | 25 (20.0%) | 39 (16.0%) | 9.30 (1), p=.335 |
| 3. | Sexual abuse | 50 (13.59%) | 23 (18.4%) | 27 (11.1%) | 3.85 (1), p=.049* |
| 4. | Emotional neglect | 85 (23.10%) | 26 (20.8%) | 59 (24.3%) | .53 (1), p=.464 |
| 5. | Physical neglect | 12 (3.26%) | 6 (4.8%) | 6 (2.5%) | 1.45 (1), p=.227 |
| 6. | Mother treated violently | 33 (8.97%) | 9 (7.2%) | 24 (9.9%) | .68(1), p=.407 |
| 7. | Substance abuse in household | 12 (3.26%) | 7 (5.6%) | 5 (2.1%) | 3.28(1), p=.070 |
| 8. | Mental illness in household | 53 (14.40%) | 20 (24.0%) | 33 (13.6%) | 6.50 (1), p=.011 * |
| 9. | Household member going to prison | 20 (5.43%) | 10 (8.0%) | 10 (4.1%) | 2.54 (1), p=.11 |
| 10. | Parental separation or divorce | 21(5.71%) | 4 (3.2%) | 17 (7.0%) | .21(1), p=.141 |

Note: *. =p <.05 level. **. =p<.01 level. ***. p= <.001 level.

**Table 3. Multinomial logistic regression results for predictors of self-reported physical health among medical students of a public sector medical university.**

| Level of physical health | Predictor | P value | Adjusted Odds Ratio | 95% confidence interval | |
|---|---|---|---|---|---|
| | | | | Lower Bound | Upper Bound |
| Poor | Intercept | .940 | | | |
| | ACE score | .353 | 1.082 | .916 | 1.277 |
| | CPF score | .304 | .815 | .551 | 1.204 |
| | Sex | | | | |
| | Male | .754 | .871 | .365 | 2.074 |
| | Female (Reference) | | 1 | | |
| Fair | Intercept | .021* | | | |
| | ACE score | .760 | .979 | .854 | 1.122 |
| | CPF score | .325 | 1.160 | .863 | 1.560 |
| | Sex | | | | |
| | Male | .493 | .770 | .364 | 1.627 |
| | Female (Reference) | | 1 | | |
| Good | Intercept | .802 | | | |
| | ACE score | .748 | 1.029 | .864 | 1.225 |
| | CPF score | .015* | 1.516 | 1.085 | 2.120 |
| | Sex | | | | |
| | Male | .012* | .249 | .084 | .735 |
| | Female (Reference) | | 1 | | |
| Excellent | Intercept | .872 | | | |
| | ACE score | .686 | .964 | .804 | 1.154 |
| | CPF score | .005* | 1.614 | 1.151 | 2.263 |
| | Sex | | | | |
| | Male | .160 | .464 | .159 | 1.354 |
| | Female (Reference) | | 1 | | |

Note: Dependent variable: Self-reported physical health. Predictor: ACE score, CPF score, and sex n= 368 medical students. *. =p <.05 level. **. =p<.01 level. ***. p= <.001 level.

was a significant predictor of good and excellent levels of physical health with adjusted odds ratio 1.516 (95% CI 1.085–2.120, p <.01) and 1.61 (95% CI 1.151–2.263, p <.001). Whereas male sex was also a significant predictor of physical health only for moderate levels with an adjusted odds ratio of.249 (95% CI .084–.735, p <.05). Results of multinomial regression are shown in Table 3.

We conducted multinomial logistic regression for self-reported physical and mental health. The model included ACE score, CPF score, mother's education, and sex as predictors for physical health. Overall, the model was a significant fit for predicting self-reported physical health ($\chi 2(24) = 59.739$, p-value, <.001) and explained a moderate degree of variance ($R^2$=.154). A higher ACE score was a significant predictor of mental health at fair, good, and excellent levels. The adjusted odds ratios are .767 (95% CI .624–.944, p <.05),.746 (95% CI .604–.922, p<.01) and .746 (95% CI .604–.922, p <.01). CPF score was a significant predictor of good and excellent mental health with adjusted odds ratio of 1.631 (95% CI 1.034–2.573, p <.05) and 1.904 (95% CI 1.198–3.026, p <.01) respectively). Whereas male sex was also a significant predictor of physical health only for excellent levels with an adjusted odds ratio of.258 (95% CI .090–.743, p <.05). Results of multinomial regression are shown in Table 4.

## Discussion

Our study highlights the association of ACE and CPF on the physical and mental health of medical students. Over the years there has been an increasing concern among academics and medical researchers on the mental and physical health of medical students [12]. There is a wealth of published research from Pakistan that reports the frequency of depressive and anxiety symptoms, quality of life, stress level, or coping mechanisms among medical students [13–16]. However, these researches have not explored causal factors from early life explaining psychiatric and physical health morbidity among medical students. To our knowledge, our study is the first study from Pakistan that explores the role of adverse childhood experiences and childhood protective factors and their impact on present health status among medical students. The results of our study show that there were no statistical differences between median ACE, CPF scores, and self-reported physical and mental health. On ten categories of ACE, there were statistically significant associations between male sex and sexual abuse as well as living with a person with a mental health condition in the household. Ansar et al., have explored ACE among medical and sociology students in Peshawar Pakistan. Emotional abuse was noted to be the common adverse childhood experience which was reported by 27.25% of participants. This percentage is similar to that found in our study. Ansar et al. reported physical abuse as the second most common ACE (17%) followed by sexual abuse (15.8%) [17]. In our study, about 18.4% of male medical students reported sexual abuse compared with 11.1% of female medical students. However, results should be interpreted with caution as they may be subjected to reporting bias. Research shows that the concept of 'honor' poses barriers for women in South Asia reporting sexual violence [18]. Ashraf et al., have conducted a study on child abuse in schools in Pakistan and have reported no significant differences in childhood sexual abuse among male and female students [19]. In another study from university students, childhood sexual abuse was reported by 39% of females and 44% of male students [20]. Sciolla et al., have reported that the most common ACE reported by male medical students was living with someone with a mental health condition (20%) followed by emotional and sexual abuse, living with someone with substance use, and parental separation (10% for each of these categories). While among female medical students living with someone with a mental health condition was 36% which was much higher than that reported in our study (13.6%) [10]. In our study of various domains of ACE, male medical students had a higher percentage suffering from all

**Table 4. Multinomial logistic regression results for predictors of mental health among medical students of a public sector medical university.**

| Level of mental health | Predictor | P value | Adjusted odds ratio | 95% confidence interval | |
|---|---|---|---|---|---|
| | | | | Lower Bound | Upper Bound |
| Poor | Intercept | .160 | | | |
| | ACE score | .078 | .827 | .669 | 1.022 |
| | CPF score | .130 | 1.426 | .901 | 2.258 |
| | Mothers' education | | | | |
| | Not educated | .342 | 3.353 | .277 | 40.593 |
| | Up to matric or religious education | .400 | 2.026 | .391 | 10.492 |
| | Intermediate or higher education but not in professional disciplines | .062 | 3.315 | .940 | 11.689 |
| | Professional education (Reference) | | 1 | | |
| | Sex | | | | |
| | Male | .628 | .805 | .334 | 1.938 |
| | Female (Reference) | | 1 | | |
| Fair | Intercept | .006** | | | |
| | ACE score | .012* | .767 | .624 | .944 |
| | CPF score | .107 | 1.451 | .922 | 2.283 |
| | Mothers' education | | | | |
| | Not educated | .511 | .407 | .028 | 5.949 |
| | Up to matric or religious education | .640 | 1.422 | .325 | 6.215 |
| | Intermediate or higher education but not in professional disciplines | .492 | 1.472 | .488 | 4.440 |
| | Professional education (Reference) | | 1 | | |
| | Sex | | | | |
| | Male | .332 | .654 | .277 | 1.543 |
| | Female | | 1 | | |
| Good | Intercept | .019* | | | |
| | ACE score | .007** | .746 | .604 | .922 |
| | CPF score | .035* | 1.631 | 1.034 | 2.573 |
| | Mothers' education | | | | |
| | Not educated | .579 | 2.060 | .161 | 26.388 |
| | Up to matric or religious education | .290 | 2.376 | .479 | 11.792 |
| | Intermediate or higher education but not in professional disciplines | .198 | 2.275 | .651 | 7.948 |
| | Professional education (Reference) | | 1 | | |
| | Sex | .079 | .440 | .176 | 1.101 |
| | Male | | | | |
| | Female (Reference) | | 1 | | |
| Excellent | Intercept | .010** | | | |
| | ACE score | .003** | .717 | .575 | .893 |
| | CPF score | .006** | 1.904 | 1.198 | 3.026 |
| | Mothers' education | | | | |
| | Not educated | .993 | .987 | .066 | 14.708 |
| | Up to matric or religious education | .948 | 1.057 | .196 | 5.692 |
| | Intermediate or higher education but not in professional disciplines | .868 | 1.115 | .309 | 4.020 |
| | Professional education (Reference) | | 1 | | |
| | Sex | | | | |
| | Male | .012* | .258 | .090 | .743 |
| | Female (Reference) | | 1 | | |

Note: Dependent variable: Self-reported physical health. Predictor: ACE score, CPF score, and sex, mother's education n= 368 medical students. *. =p <.05 level. **. =p<.01 level. ***. p= <.001 level.

types of ACE categories except emotional neglect, being treated with violence by their mother, and separation or divorce between parents, which were more common among female medical students. This is contrary to findings reported from Saudia Arabia, Iran, USA, and India where ACE for all categories were more common among females [21–24]. However, in these studies, ACE were reported by the general population and not medical students. It is not clear whether the gender differences seen in our study are due to population attributes or cultural differences.

We also explored the impact of ACE and CPF on self-reported physical and mental health. ACE was not found to be a significant predictor of self-reported across various categories of physical health. Monnat et al. explored the impact of ACE on 52,250 adults from 18-64 years of age. The authors report that those with higher ACE were less likely to have health checkups in the past two years, have health insurance, and reported engaging exercise over the last month. A meta-analysis published by Hughes and colleagues reported a weak to modest effect of ACE on physical health. The results report an odds ratio of 1.25 for physical inactivity, 1.12 for overweight or obesity, and 1.52 for diabetes. In contrast to ACE, CPF was a significant predictor of self-reported physical health but only in good and excellent categories, though the odds ratio was of modest effect [3].

In terms of the impact of ACE and CPF on self-reported mental health, higher ACE was associated with lesser odds of reporting fair, good, and excellent mental health. Whereas a higher CPF score was associated with greater odds of reporting mental health in good and excellent categories. Mothers' education was not a significant predictor of mental health at any level. Whereas male medical students were less likely to report excellent mental health as opposed to very poor mental health while controlling for ACE score. Hughes et al. reported that the association between mental health and ACE was stronger than between physical health and ACE. According to the study, a higher ACE score was associated with an odds ratio of 30.14 for suicidal attempts, 10.22 for problematic drug use, 5.84 for problematic alcohol use, 4.40 for depression, and 3.70 for anxiety [3]. However, the meta-analysis from Hughes et al. included a diverse population and did not necessarily include medical students. There is a dearth of studies on the impact of ACE on mental health [25]. We identified another study where the impact of ACE was assessed on veterinary medical students. The study reports a threefold increase in depressive symptoms among students reporting an ACE score of four or more. While it may seem intuitive that medical students have fewer barriers to accessing mental health services, research has shown that 56% of medical students in Pakistan have a negative attitude toward seeking mental health services in Pakistan [26].

## Limitations and recommendations

Our study did not explore whether the impact of ACE is mitigated by childhood protective factors, personality attributes, resilience, or coping mechanisms. The mediating effects of these attributes are studied elsewhere in the literature and may have a potential role in preventative measures. Moreover, the cross-sectional design of the study does not establish causality. Our study included students from one medical university; thus, generalization of results could not be made. Further research from more representative samples across the country may enrich our understanding of this research area. Research on mitigating factors for ACE may highlight points of intervention for providing support services and fostering resilience among medical students experiencing a higher degree of ACE. Medical colleges should also routinely screen students for mental health issues and providers should explore ACE in appropriate settings in a sensitive manner among those reporting poor mental health. Given the stigma associated with childhood adversities, universities may include awareness and de-stigmatization campaigns so that students may open up about their experiences and seek mental health services.

## Conclusion

The impact of ACE and CPF on self-reported physical and mental health is underexplored among medical students in the literature. Our study shows that ACE was a significant predictor of self-reported mental health but not physical health. However, CPF was a significant predictor of physical and mental health.

## Author contributions

**Conceptualization:** Usman Ali, Nazish Imran.

**Data curation:** Usman Ali, Muhammad Shaheryar Khan.

**Formal analysis:** Usman Ali, Nazish Imran, Muhammad Shaheryar Khan.

**Investigation:** Usman Ali, Muhammad Shaheryar Khan.

**Methodology:** Nazish Imran.

**Project administration:** Muhammad Shaheryar Khan.

**Writing – original draft:** Usman Ali.

**Writing – review & editing:** Nazish Imran, Muhammad Shaheryar Khan.

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
