## [Decision Letter · Decision Letter 0]

7 Jan 2025

PGPH-D-24-02758

Impact of Adverse Childhood Experiences (ACE) and Childhood Protective Factors (CPF) on physical and mental health of medical students of a public sector medical university.

Dear Dr. Usman,

Thank you for submitting your manuscript to PLOS Global Public Health. After careful consideration, we feel that it has merit but does not fully meet PLOS Global Public Health’s publication criteria as it currently stands. Therefore, we invite you to submit a revised version of the manuscript that addresses the points raised during the review process.

We look forward to receiving your revised manuscript.

Kind regards,

Sualeha Siddiq Shekhani

Academic Editor

Journal Requirements:

1. Please provide additional details regarding participant consent. In the ethics statement in the Methods and online submission information, please ensure that you have specified (1) whether consent was informed and (2) what type you obtained (for instance, written or verbal, and if verbal, how it was documented and witnessed). If your study included minors, state whether you obtained consent from parents or guardians. If the need for consent was waived by the ethics committee, please include this information.

3. In the online submission form, you indicated that "Will be available one request. Shall upload on github.". 

3. Uploaded as supplementary information.

4. Please insert an Ethics Statement at the beginning of your Methods section, under a subheading 'Ethics Statement'. It must include:

1) The name(s) of the Institutional Review Board(s) or Ethics Committee(s)

2) The approval number(s), or a statement that approval was granted by the named board(s) 

3) (for human participants/donors) - A statement that formal consent was obtained (must state whether verbal/written) OR the reason consent was not obtained (e.g. anonymity). NOTE: If child participants, the statement must declare that formal consent was obtained from the parent/guardian.

5. Please provide an Author Summary. This should appear in your manuscript between the Abstract (if applicable) and the Introduction, and should be 150–200 words long. The aim should be to make your findings accessible to a wide audience that includes both scientists and non-scientists. Sample summaries can be found on our website under Submission Guidelines:

https://journals.plos.org/globalpublichealth/s/submission-guidelines#loc-parts-of-a-submission

6. We have noticed that you have cited Supporting Information files in your manuscript's Data availability on page 15. However, there are no corresponding files uploaded to the submission. Please upload them as separate files with the item type 'Supporting Information'. 

Additional Editor Comments (if provided):

This is an important and well-written study.

My concern is that it is limited to one private medical college therefore the results cannot be generalized. This should be mentioned in the Limitations section. In addition, there should be some mention of the applicability of these results from a public health perspective.

Reviewers' comments:

Reviewer's Responses to Questions

**Comments to the Author**

1. Does this manuscript meet PLOS Global Public Health’s publication criteria ? Is the manuscript technically sound, and do the data support the conclusions? The manuscript must describe methodologically and ethically rigorous research with conclusions that are appropriately drawn based on the data presented.

Reviewer #1: Yes

Reviewer #2: Yes

2. Has the statistical analysis been performed appropriately and rigorously?

Reviewer #1: Yes

Reviewer #2: Yes

3. Have the authors made all data underlying the findings in their manuscript fully available (please refer to the Data Availability Statement at the start of the manuscript PDF file)?

Reviewer #1: No

Reviewer #2: Yes

4. Is the manuscript presented in an intelligible fashion and written in standard English?

Reviewer #1: Yes

Reviewer #2: Yes

5. Review Comments to the Author

Reviewer #1: Very well written paper.

One of your findings indicates that ACE does not have a significant effect on physical health. I wonder if this has anything to do with the fact that your participants are medical students who are probably more equipped or comfortable with seeking physical health care (than mental health care) and not because ACE just does not have an effect on physical health. In which case this same finding will not be expected of a more general population or populations that are not as equipped to seek physical health care as the medical students.

You are free to include this line of thought in your manuscript if you see my point.

Reviewer #2: Authors should focus on correcting grammatical errors in their manuscripts and ensure proper punctuation placement. While statistical reporting is acceptable, authors must also verify the grammatical interpretation of their results.

6. PLOS authors have the option to publish the peer review history of their article (what does this mean? ). If published, this will include your full peer review and any attached files.

**Do you want your identity to be public for this peer review?** For information about this choice, including consent withdrawal, please see our Privacy Policy .

Reviewer #1: No

Reviewer #2: **Yes: ** EMMANUEL ABU BONSRA

---

## [Editor Report · Decision Letter 1]

30 Jan 2025

Impact of Adverse Childhood Experiences (ACE) and Childhood Protective Factors (CPF) on physical and mental health of medical students of a public sector medical university.

PGPH-D-24-02758R1

Dear Dr. Khan

We are pleased to inform you that your manuscript 'Impact of Adverse Childhood Experiences (ACE) and Childhood Protective Factors (CPF) on physical and mental health of medical students of a public sector medical university.' has been provisionally accepted for publication in PLOS Global Public Health.

Best regards,

Sualeha Siddiq Shekhani

Academic Editor